# Emergence of visually-evoked reward expectation signals in dopamine neurons via the superior colliculus in V1 lesioned monkeys

Norihiro Takakuwa[1,2,3], Rikako Kato[1,3], Peter Redgrave[4], Tadashi Isa[1,2,3]*

[1]Department of Developmental Physiology, National Institute for Physiological Sciences, Okazaki, Japan; [2]Department of Physiological Sciences, SOKENDAI, Hayama, Japan; [3]Department of Neuroscience, Graduate School of Medicine Kyoto University, Kyoto, Japan; [4]Department of Psychology, University of Sheffield, Sheffield, United Kingdom

**Abstract** Responses of midbrain dopamine (DA) neurons reflecting expected reward from sensory cues are critical for reward-based associative learning. However, critical pathways by which reward-related visual information is relayed to DA neurons remain unclear. To address this question, we investigated Pavlovian conditioning in macaque monkeys with unilateral primary visual cortex (V1) lesions (an animal model of 'blindsight'). Anticipatory licking responses to obtain juice drops were elicited in response to visual conditioned stimuli (CS) in the affected visual field. Subsequent pharmacological inactivation of the superior colliculus (SC) suppressed the anticipatory licking. Concurrent single unit recordings indicated that DA responses reflecting the reward expectation could be recorded in the absence of V1, and that these responses were also suppressed by SC inactivation. These results indicate that the subcortical visual circuit can relay reward-predicting visual information to DA neurons and integrity of the SC is necessary for visually-elicited classically conditioned responses after V1 lesion.

*For correspondence: isa. tadashi.7u@kyoto-u.ac.jp

**Competing interests:** The authors declare that no competing interests exist.

## Introduction

Adaptive behaviour in a changing environment requires that we have to learn and update associations between unconditioned rewards and punishments, and the sensory stimuli that predict them. This form of associative learning is called classical or Pavlovian conditioning (*Pavlov, 1927*). The Pavlovian paradigm has been used widely to investigate the role of midbrain dopamine (DA) neurons in associative learning (*Schultz, 1998*). Much evidence indicates that the activity of DA neurons in the substantia nigra pars compacta (SNc) makes a key contribution in associative learning, in part, by encoding reward prediction errors. A reward prediction error is a scalar signal that signifies a current event is better or worse than predicted.

In a series of pioneering experiments Schultz and colleagues (*Schultz et al., 1992*, *1997*; *Mirenowicz and Schultz, 1994*) showed that DA responses to an unpredicted reward (unconditioned stimulus; UCS), gradually transferred to an unexpected predicting conditioned stimulus (CS). If a predicting CS was presented but subsequent reward delivery was withheld, DA neurons would pause briefly at the expected time of reward delivery (*Schultz et al., 1997*). These bidirectional sensory responses of DA neurons to events that were better or worse than expected led to the formulation of the reward prediction error hypothesis of DA signaling (*Montague et al., 1996*; *Schultz, 1998*). Subsequent experiments have confirmed that phasic DA responses are sensitive to

**eLife digest** To survive and thrive, animals must learn to approach cues in their environment that are likely to lead to a desirable outcome and avoid those that might lead them to harm. A group of brain regions known as the midbrain dopamine system helps many animals to achieve this. Dopamine is the brain's reward signal. Cues that predict rewards, such as the sight or smell of food, activate midbrain dopamine neurons. However, the details of this process remained unclear.

Takakuwa et al. have now examined how visual information that signals reward reaches the midbrain dopamine neurons. The anatomy of the visual system suggests two main possibilities. Information may travel directly from the eyes to an area of the midbrain called the superior colliculus, and then onto the dopamine neurons. Alternatively, information may travel to the midbrain indirectly via a pathway that includes additional processing in the brain's outer layer, the visual cortex.

To distinguish between these routes, Takakuwa et al. studied monkeys in which the indirect pathway via the visual cortex had been damaged. Some people with damage to this pathway have a disorder called blindsight. They are able to detect the movement or location of stimuli, but they cannot consciously see those stimuli. The monkeys with damage to visual cortex were able to learn that an image on a screen predicted the delivery of fruit juice. After repeated trials, the monkeys began to lick the spout dispensing the juice whenever the image appeared, even if no juice was delivered. The monkeys' midbrain dopamine neurons also sent more signals in response to the images, and showed greater activity when the images predicted large rewards than small ones. Takakuwa et al. next inactivated the superior colliculus with a drug and showed that this prevented both the licking behavior and the increased signaling.

Together the findings show that visual information about potential rewards can reach midbrain dopamine neurons via a direct route through the superior colliculus, without needing to pass via the visual cortex. The next step is to determine how and when the visual cortex may get involved in this process to help animals maximize rewards.

reward magnitude (*Tobler et al., 2005*), reward probability (*Fiorillo et al., 2003*; *Nakahara et al., 2004*; *Matsumoto and Hikosaka, 2009*) and reward delay (*Kobayashi and Schultz, 2008*; *Fiorillo et al., 2008*).

It has been shown that short latency phasic responses can be elicited in DA neurons by unexpected rewards (*Schultz, 1998*; *Fiorillo, 2013*) or conditioned stimuli that predict future reward (*Matsumoto and Hikosaka, 2009*, *2007*; *Eshel et al., 2015*). A critical feature of these early experiments was that the latency of sensory (usually visually) elicited DA responses was typically 100 ms or less following stimulus onset. This raised the question of by which route(s) is the visual information for reward expectation relayed to DA neurons in the ventral midbrain (*Redgrave et al., 1999*). In a series of investigations, a novel projection from the subcortical midbrain superior colliculus (SC) directly to the midbrain DA neurons was demonstrated in rat (*Comoli et al., 2003*), cat (*McHaffie et al., 2006*) and monkey (*May et al., 2009*). The SC is an evolutionary archaic visual structure in the vertebrate brain that receives direct input from retinal ganglion cells (*Perry et al., 1984*), and is especially sensitive to unexpected luminance changes (*Boehnke and Munoz, 2008*). A later study (*Dommett et al., 2005*) confirmed that the retino-tecto-nigral projections were involved in the short-latency phasic activation and release of DA in the basal ganglia following a transient light-flash. However, this investigation was conducted in anaesthetized rodents, and it remains to be determined whether the SC can play a critical role in the short-latency CS-elicited activation of DA neurons and conditioned responses in awake behaving non-human primates.

During the evolutionary expansion of the cerebral cortex, the relative importance of the geniculostriate projection to primary visual cortex (V1) for visual perception increased (*Livingstone and Hubel, 1988*). This development offered a further potential route via V1, by which visual information for reward expectation might be relayed to ventral midbrain DA neurons. Therefore, the specific purpose of the present study was to investigate whether the subcortical visual pathway via the SC can mediate the afferent visual CS signal in the Pavlovian conditioning paradigm and activate DA

neurons at short-latency in primates. To do this, we used monkeys that had a unilateral lesion of cortical area, V1. This preparation in which primary cortical visual processing was disabled was used to isolate the contribution of the SC that remained intact on the V1 lesioned side. After V1 damage, visual awareness is impaired in the lesion-affected visual field (*Cowey and Stoerig, 1995*; *Yoshida and Isa, 2015*). However, from both human (*Poppel et al., 1973*) and animal studies (*Cowey and Stoerig, 1995*; *Yoshida and Isa, 2015*) it is known that a transient visual stimulus presented in the lesion-affected visual field can trigger a range of behavioural responses, in the apparent absence of subjective awareness. This phenomenon has been called 'blindsight', where many of the residual visual competences are thought to be mediated by the SC (*Mohler and Wurtz, 1977*). Consequently, we have made use of animals that were used previously to characterize the phenomenon of 'blindsight'; they have abilities to make saccadic eye movements to a visual target presented in the lesion-affected visual field (*Yoshida et al., 2008*), despite their awareness to the visual target was impaired like human blindsight patient (*Yoshida and Isa, 2015*). This animal model enabled us to test whether the intact subcortical visual circuitry in this preparation can support visual Pavlovian conditioning and short-latency activation of DA neurons (*Schultz, 1998*).

The purpose of this study was, therefore, to test whether unilaterally V1-lesioned monkeys could associate reward-predicting visual cues with subsequent reward (Pavlovian conditioning), and whether visual CSs could activate midbrain DA neurons. To verify the role of subcortical visual processing, neural activity in the SC was suppressed with local injections of a pharmacological agent.

## Results

### V1 lesion

The right V1 of monkey K and U, and left V1 of monkey T was surgically removed by aspiration, 46, 44 and 6 months before the present experiments, respectively. The lesion area was confirmed by MR images and the range of the lesion-affected visual field was confirmed by increased threshold for detecting saccadic targets at the beginning of the present experiments (*Figure 1A*, *Figure 1—figure supplement 1*). We presented targets at possible positions which covered the whole contralesional visual field (monkey K; 3 directions × 3 eccentricities, monkey U; 5 directions × 4 eccentricities, monkey U; 5 directions × 3 eccentricities) and luminance contrast sensitivity of all targets to induce saccadic eye movements clearly decreased in affected visual field (*Figure 1—figure supplement 1C*). The visual deficits caused by these lesions was similar to the animals which were reported previously (*Yoshida et al., 2008*). These results indicated that the V1 lesion affected most of the contralesional visual field, at least from 5° to 15° eccentricities. Visual input pathways from retina can be classified into two major pathways; one is cortical pathways via LGN and V1, the other is subcortical pathways via the SC. The monkeys with unilateral V1 lesion were used to investigate abilities of the subcortical visual pathways through the SC (*Mohler and Wurtz, 1977*; *Kato et al., 2011*; *Takaura et al., 2011*). In this study, the V1 lesion allowed us to assess contribution of visual information via the SC to support visual classical conditioning and to evoke phasic DA responses following the presentation of conditioned stimuli.

### Pavlovian conditioning

As a first step we investigated whether monkeys K and U, both with unilateral lesions of V1, could learn the association between a visual CS and subsequent reward when the CS was presented in the lesion-affected 'blind' field (*Figure 1A*). In this part of the study we presented two visual CSs; one predicted a large reward (LR = 0.17 ml) delivered during the CS presentation (1.3 s from CS onset), whereas the other predicted a small reward (SR = 0.06 ml) delivered 1.5 s after the CS offset. The two CSs could be discriminated by their location relative to central fixation point (upper or lower visual field, *Figure 1B*). On separate days the CSs were presented to the lesion-affected or intact visual fields.

After 12 days of having the CSs predict juice delivery (approximately 200 trials/day), conditioned anticipatory licking was induced by both LR-CS and SR-CS (*Figure 1C*, *Figure 1—figure supplement 2A*). The conditioned licking rate during the CS presentation was significantly higher in LR trials than in SR trials (15 sessions in monkey K and 16 sessions in monkey U, *Figure 1D*, $\alpha$ <0.05, Wilcoxon signed-ranks test). In addition, the conditioned responses elicited by CSs presented to either the

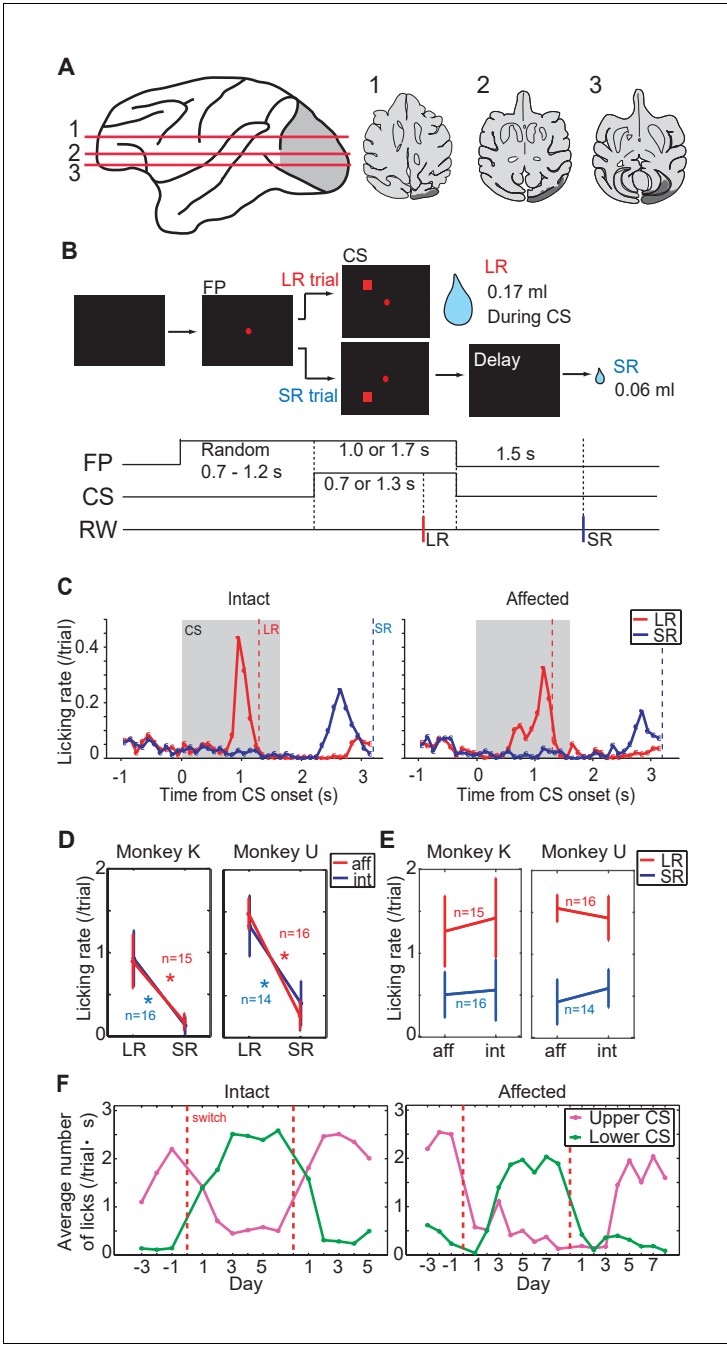

**Figure 1.** Pavlovian conditioning in V1 lesioned monkeys. (**A**) Left: lesion area (depicted in gray) on the whole brain image. Red lines (1 - 3) indicate dorso-ventral levels of horizontal slices shown on the right. Right: lesion area in monkey K (depicted in gray) is overlaid as black areas on axial slices traced from MR images. (**B**) Design of Pavlovian conditioning task in this study. Monkeys were required to fixate a central fixation point (FP) until CS offset. LR (large reward) and SR (small reward) trials were given at random order. In this task, LR was delivered during CS presentation, and SR was delivered after 1.5 s from CS offset. Abbreviations; RW (reward). (**C**) Licking rates aligned at the CS onset (monkey K). CSs were presented to intact visual field (left panel) and to lesion-affected visual field (right panel). Red and blue lines indicate licking rates during LR and SR trials, respectively. Gray hatched area indicates CS presentation period. Red and blue vertical dashed lines indicate time of reward delivery in the LR and SR trials, respectively. (**D**) Licking rates during CS presentation were compared between LR and SR trials in monkey K (left) and U (right). The CSs were presented either to the intact (int, blue lines) or lesion-affected (aff, red lines) hemifield. * = significant difference (monkey K: p=6.1 × $10^{-5}$ (aff), p=4.3 × $10^{-4}$ (int), monkey U: p=4.3 × $10^{-4}$ (aff), p=1.2 × $10^{-4}$ (int), Wilcoxon signed-ranks test, α <0.05). (**E**) Licking rates during CS
*Figure 1 continued on next page*

*Figure 1 continued*

presentation were compared between CS presented to lesion-affected and that to intact visual field in monkey K (left) and U (right). There was no significant difference in the licking rates both in LR and SR trials. monkey K: p=0.33 (LR), p=0.63 (SL), monkey U: p=0.16 (LR), p=0.084 (SL), two sample t-test with Welch's correction, α <0.05. (F) Reversal learning; the effect of switching the CS assignment on licking rates in the intact and affected fields in monkey K. Licking rates during CS presentation to upper (magenta) or lower (green) visual field were plotted for individual days. CS positions were switched on the day indicated by the vertical red dashed lines.

The following figure supplements are available for figure 1:

**Figure supplement 1.** Unilateral V1 lesion.
**Figure supplement 2.** Pavlovian conditioning in monkey U.

intact or lesion-affected visual fields were not reliably different (*Figure 1E*), (α <0.05, two sample t-test with Welch's correction). These results show that a visual cue presented in the V1 lesion-affected hemi-field can act as an effective CS in a Pavlovian conditioning task. Moreover, the monkeys were able to discriminate successfully between the difference in the magnitude and timing of reward predicted by CSs according to where they were presented in the lesion-affected hemi-field.

## Reversal learning

To test the flexibility of associative learning and to exclude the possibility that the discriminability of the LR- and SR-CSs was simply determined by their respective locations, the upper and lower positions on the screen where the LR-CS and SR-CS appeared were switched (*Figure 1F*, *Figure 1—figure supplement 2B*). After the switching, the high conditioned licking rate gradually changed to follow the new LR-CS, again irrespective of whether the CSs were presented in both intact and lesion-affected visual fields (*Figure 1F*). After the successful reversal, the LR- and SR-CSs were switched back to their original assignment. At which point the conditioned responses switched back to follow the newly assigned LR-CS. These results indicate that monkeys can flexibly associate the locations of the visual CSs and the reward predicted by them even without V1.

## Muscimol injection

To investigate whether visual processing in the SC was responsible for the expression of visually-evoked conditioned responses when the CSs were presented to the V1 lesion affected side, the GABA agonist muscimol (0.5 µL; 1 µg/µL concentration at a rate of 1 µL/15 s) was injected into the ipsi-lesional SC of monkeys K and T. Thus, before the muscimol injection, neural activity of the SC was recorded, and the location of neurons responsive to LR-CS was identified on SCs retinotopic map. Muscimol was then injected into this location (*Figure 2A*). The suppressive effect of the muscimol injection was confirmed by showing that the monkey failed to make saccades to the LR-CS location as previously shown for the blindsight monkeys by *Kato et al. (2011)* (*Figure 2B*; see disappearance of saccades to the left-upward target).

Also, before the muscimol injection, anticipatory licking evoked by the LR-CS presentation (0–0.7 or 1.3 ms) served as a baseline control in our Pavlovian conditioning task (*Figure 2C* left). Immediately following the muscimol injection the monkeys continued to perform the LR-CS evoked conditioned anticipatory licking. However, over the next 20–30 min the normal conditioned response (anticipatory licking) gradually disappeared (*Figure 2C* right). At which point, two new patterns of behaviour were observed: (i) in the case of monkey T (*Figure 2C* right), all anticipatory response was abolished and licking appeared only after the juice reward was delivered; and (ii) for monkey K anticipatory licking was evoked shortly after the onset of both the LR-CS and SR-CS (*Figure 2—figure supplement 1*. In other words, the animal's ability to discriminate between the CSs on the basis of position within the visual field was lost.

Muscimol injections were administered in 13 experiments (monkey K: 9 experiments, monkey T: 4 experiments). To assess the effect of the SC inactivation the difference between the licking rate during CS presentation was compared for LR and SR trials. Before the SC inactivation (control), monkeys licked a reward spout more frequently during CS period in LR trials than in SR trials in all sessions.

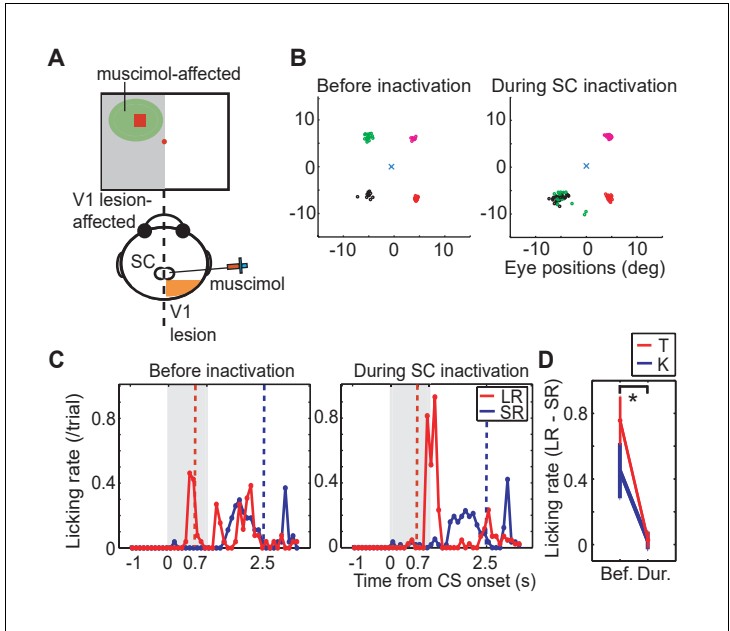

**Figure 2.** Effect of SC inactivation on conditioned behaviors. (**A**) A scheme of the SC inactivation experiments. Muscimol was injected into the point on the ipsi-lesional SC map representing the location of LR-CS in the visual field. (**B**) End points of saccadic eye movements before and after the SC inactivation (left and right panel). The position of central fixation point is indicated by a blue cross. Circles indicate end points of visually guided saccades, and their colors indicate location of saccadic targets in individual quadrants. Impairment of saccades toward the upper-left target (green) indicates that muscimol effectively suppressed the neuronal activity at the injection site. (**C**) Licking rates in a daily session before (left panel) and after SC inactivation (right panel) in monkey T. The licking rates are plotted in the same manner as *Figure 1C*. Red and blue lines indicate the licking rates during the LR and SR trials, respectively. Gray hatched area indicates the CS presentation period. (**D**) Licking rate during 0.7 s from the CS onset in the SR trials are subtracted from licking rate in LR trials in monkey K (blue line, N = 9) and T (red line, N = 4). The vertical lines indicate the SEM. Bef.: before inactivation, Dur: during inactivation. (p=2.4 × 10$^{-4}$, Wilcoxon signed-ranks test, α <0.05).

The following figure supplement is available for figure 2:

**Figure supplement 1.** Effect of SC inactivation on conditioned behavior in monkey K.

---

The difference of the licking rate between in LR trials and in SR trials was diminished after SC inactivation (Wilcoxon signed-ranks test; p<0.001). During the SC inactivation, the difference of licking rate was not significantly different from zero (one-sample t-test; p>0.005). The results were consistent in all sessions of both monkeys (*Figure 2D*).

These results indicate that the visual processing signifying CS onset by the SC on the V1 lesion-affected side was essential for a previously established conditioned response to be expressed in our Pavlovian conditioning task.

## Responses of DA neurons to visual conditioned stimuli

It has been reported widely that dopamine neurons are phasically activated by unpredicted conditioned stimuli in Pavlovian tasks (*Schultz, 1998*). The purpose of the next phase of our study was, therefore, to investigate whether a visual CS presented to the V1 lesion-affected visual field had the capacity to evoke a phasic response in ipsilateral DA neurons in the current Pavlovian conditioning task. Monkeys K and T were used for these experiments.

Neurons conforming to the electrophysiological criteria established for identifying putative DA neurons were recorded in the ventral midbrain. The neurons included in our sample therefore had low baseline firing rates (<10 Hz), and broad spike-widths (>0.45 ms between the first negative peak and next positive peak) (*Figure 3B,C*). The location of recorded neurons was later confirmed by

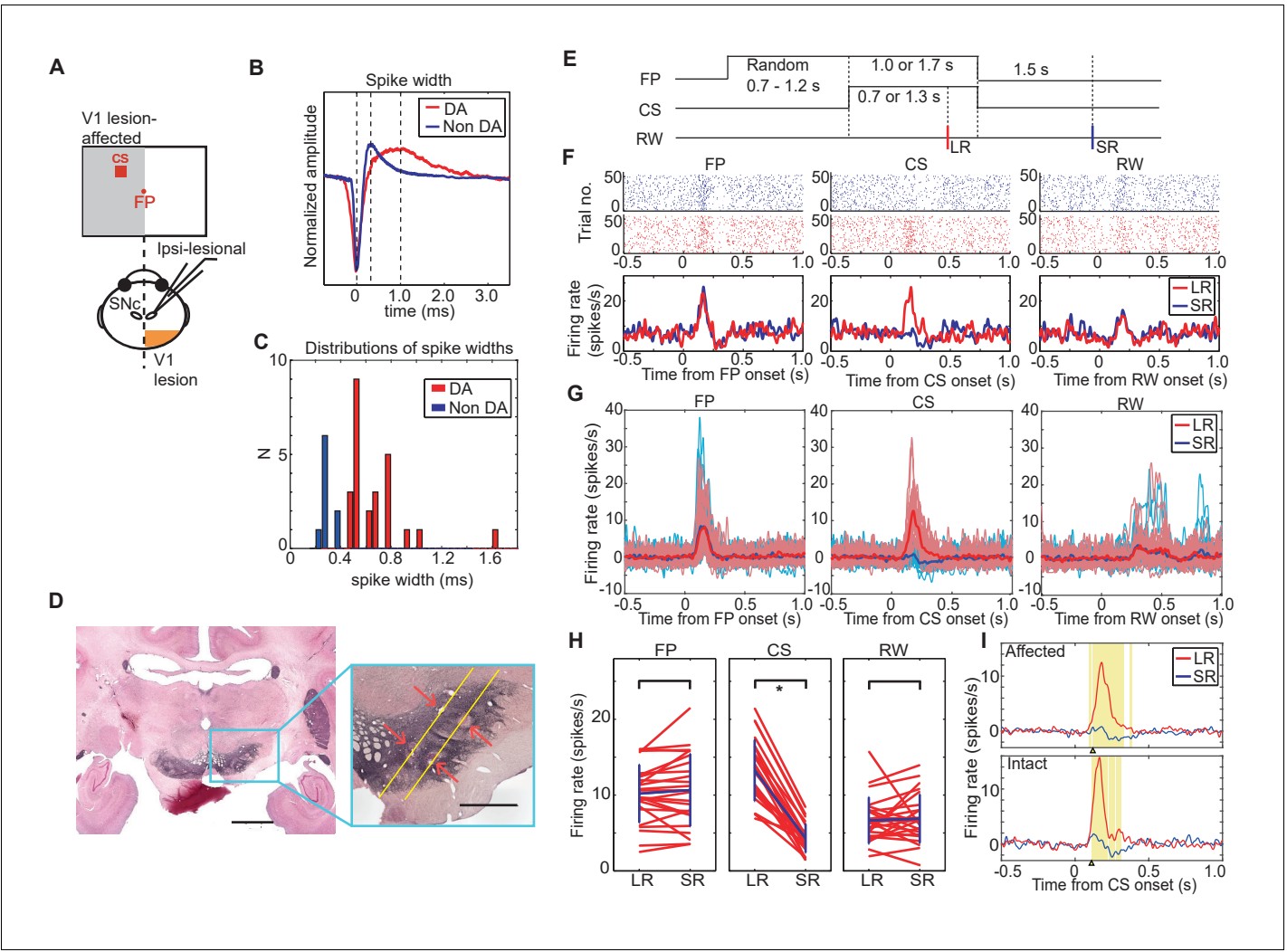

**Figure 3.** DA neuron responses during Pavlovian conditioning task. (**A**) Schematic drawing of the experimental design for recording DA neuron activity in the monkey with unilateral V1 lesion. (**B**) Averaged spike waveforms of a presumed DA neuron in SNc and a non-DA neuron in the SNr. Amplitude of these spikes are normalized. Spike width was defined as the time between the first negative peak and second positive peak. (**C**) Histogram of the spike width. Red bars indicate the DA neurons and blue bars indicates the SNr neurons. (**D**) Left; a low magnification view of the SNc and surrounding structures stained with anti-TH immunohistochemistry. Scale bar = 5.0 mm. Right; a high magnification view of the area indicated by a blue square. Red arrows indicate locations of electrolytic markings. Scale bar = 2.0 mm. (**E**) Time course of the Pavlovian conditioning task (the same as *Figure 1B*). (**F**) A typical DA neuron activity in V1 lesioned monkeys. Raster plots of a DA neuron from LR (red) and SR (blue) trials were sorted and shown on the top, receptively. The first trial was plotted at the bottom of the raster plot and the last trial was plotted at the top. Red and blue lines indicate average firing rates during LR and SR trials, respectively. These plots were aligned at the FP onset, CS onset, and RW delivery (left, middle and right panels, respectively). (**G**) Responses of all recorded DA neurons to FP, CS and RW (left, middle and right panels) are superimposed. A thick red line in each panel is the averaged firing rate of DA neurons in LR trials, and a thick blue line is the averaged firing rate in SR trials. Thin lines behind the averaged lines are the averaged responses of individual neurons in LR trials (red) and in SR trials (blue), respectively. (**H**) Firing rates of individual DA neurons within the time windows (100–300 ms from FP and CS or 150–350 ms from RW; left, middle and right panels). Blue lines indicate the average of all the neurons and SD of the firing rate in LR trials and in SR trials. * = significant difference (N = 24, p=0.82 (FP), p=1.1 × 10$^{-7}$ (CS), p=0.27 (RW), Wilcoxon signed-ranks test, α <0.05). (**I**) The yellow background in the figures shows the period during which the responses to LR-CS and SR-CS were significantly different more than 15 ms (N = 24 in affected, N = 16 in intact, two-sided sign test, α <0.05). The two panels show averaged DA responses to CSs presented to the lesion-affected visual field (upper panel), and to the visual field (lower panel). Arrows under each figure indicate the earliest points where the LR and SR responses can be reliably discriminated for more than 50 ms (122 ms in the lesion-affected visual field, and 112 ms in intact visual field).

The following figure supplement is available for figure 3:

**Figure supplement 1.** Comparing DA responses to CSs in lesion-affected and intact visual field.

identifying the site of small lesions made at some of the recording sites in tissue immunostained for tyrosine hydroxylase (*Figure 3D*).

Typical responses of putative DA neurons in our Pavlovian task are shown for a single case (*Figure 3F*), and for the population of recorded neurons (n = 24) (*Figure 3G*). First, because of its task-relevance and unpredictability, putative DA neurons were activated robustly by the onset of the fixation point. However, this response was similar in LR trials and SR trials (left-hand panels of *Figure 3F and G*) because at the time the fixation point was presented the magnitude and timing of reward predicted by the upcoming CS was unknown. Subsequently, when the temporally uncertain CSs were presented, a clear difference in the putative dopamine response was evident between the LR and SR trials – a reliably larger response was evoked by the LR-CS (central panels of *Figure 3F and G*). In this case, responses to predicted presentations of the juice reward were unreliable and significantly weaker than responses evoked either by the FP or CSs (right-hand panels of *Figure 3F and G*).

Confirmation of the above findings for the population of DA neurons (n = 24) is illustrated in *Figure 3H*. In these figures, firing rate of these responses in a selected time window (FP, CS: 0.1 s – 0.3 s from the onset, RW: 0.15 s – 0.35 s from the delivery) was compared between the LR and SR trials. The left-hand panel shows that there was no reliable difference between the putative dopamine responses evoked by FP presentation in LR and SR trials (Wilcoxon signed-ranks test). However, the LR-CS elicited a significantly larger responses compared with those evoked by the SR-CS (central panel *Figure 3H*). These responses were not strongly affected by the V1 lesion. Firing rate of the responses to CSs presented into lesion affected and intact visual fields were not significantly different (*Figure 3—figure supplement 1*). Finally, there were no reliable differences in the responses evoked by the onset of the predicted LR or SR (right-hand panel *Figure 3H*).

The overall mean response latency was 107 ms while the latencies of the individual neurons were distributed between 60 to 160 ms after the LR-CS onset (latency = the time when the neural response rate exceeded 2SD of their baseline activity). We calculated the earliest time points when difference between responses to LR-CS and SR-CS was observed. The earliest time points when response differentiation lasting more than 15 ms started was 122 ms from the CS onset in lesion-affected visual field, and 112 ms in intact visual field (*Figure 3I*). This result indicates that the latency of the reward discrimination by DA neurons was minimally affected by the absence of V1.

These results showed in the absence of V1, that temporally unpredicted visual CSs were able to elicit typical short latency and short duration phasic responses in ventral midbrain neurons, presumed to be dopaminergic. These neurons could discriminate the LR-CS and SR-CS, based on the location of their presentation within the lesion-affected visual field. These results indicate that the residual early visual structures (most likely the midbrain SC) retained the capacity to evoke differential phasic DA responses informed by the reward expected from CS. The final phase of our study sought to test the contribution of the SC.

## CS evoked responses during SC inactivation

To test whether the transmission of visual signals via the SC was responsible for CS-evoked phasic DA responses, muscimol was injected into the ipsi-lesional SC (*Figure 4E*). Thus, after the collection of control data on visually guided saccadic task and on Pavlovian conditioning task, baseline records of the responses of the DA neurons to the presentation of the fixation point, CS and reward were recorded. When all was done, muscimol was injected into the appropriate location of the SC (see above) and DA responses to the same sensory events were reassessed. Thus, the activity of a single DA neuron was recorded both before and after the muscimol injection. To ensure that the same recorded neuron was maintained throughout the session (that is, for approximately 1.5 hr), its waveform was carefully monitored. Only when the DA waveforms remained constant before, after muscimol injection were the data included in our sample (*Figure 4—figure supplement 1*).

The responses of a typical DA neuron are illustrated in *Figure 4A and B*. Before collicular inactivation (*Figure 4A*) the DA responses to the task-related stimuli were similar to those observed in previous experiments (see above – *Figure 3F and G*). After the injection of muscimol, when the relevant SC was inactivated, the robust response evoked by the FP was largely unaffected (compare *Figure 4A and B* (left-hand panels), *Figure 4C* left and 4D left, Wilcoxon test, not significantly different). After the muscimol injection the response of the recorded neuron to presentation of the LR-CS was retained for a short while (central panels *Figure 4B*). However, after a few trials the drug action

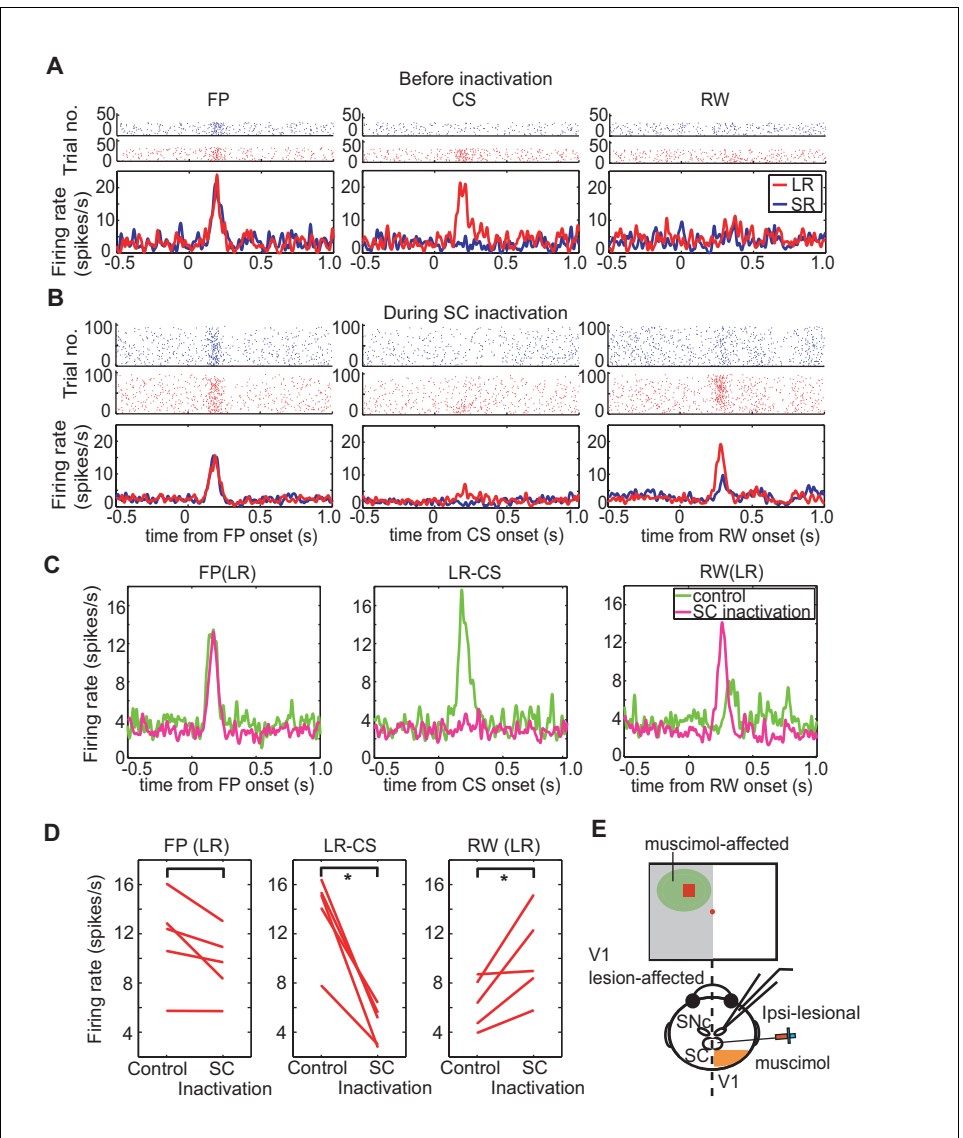

**Figure 4.** Effect of SC inactivation on cue-responses in DA neurons. (**A**) Activity of DA neurons before SC inactivation. Raster plots and firing rates plotted in the same manner as *Figure 3F*. These plots were aligned at FP onset, at CS onset, and at RW delivery (left, middle and right panels, respectively). (**B**) Activity of DA neurons during SC inactivation After the SC inactivation, the responses to the FP were unchanged (left), those to the LR-CS (middle) disappeared and those to RW (right) increased. (**C**) Population average of DA neuron responses (N = 5) in LR trials before (green) and during SC inactivation (magenta). These activities were aligned at FP onset, at CS onset and at RW delivery, respectively (left, middle and right panels). (**D**) Firing rates of DA neurons in LR trials within different time windows (100–300 ms from FP and CS or 150–350 ms from RW; left, middle and right panels, respectively) before and during SC inactivation. These time windows are the same as those in *Figure 3H*. * = significant difference (N = 5, p=0.067 (FP), p=0.0025 (CS), p=0.043 (RW), one sample t-test, α <0.05). (**E**) A schematic drawing of the experimental setup for the DA neuron recording and SC inactivation. Ipsi-lesional SC was inactivated. The neural activity was recorded from the ipsi-lesional SNc.

The following figure supplements are available for figure 4:

**Figure supplement 1.** Spike waveforms of a DA neuron during a daily session.

**Figure supplement 2.** firing rate of responses to SR-CS.

became apparent, and the CS-evoked response was almost completely abolished (central panel *Figure 4B and C*). It is also significant that in these early trials, when the reward delivery was still predicted by visual input from the SC, reward presentation failed to evoke a phasic DA response. However, as the colliculus became fully inhibited, the now unpredicted presentation of the reward evoked a robust phasic response, which in this case was clearly dependent on the magnitude and timing of reward predicted by the CS. This pattern of response was consistent in all recorded neurons (*Figure 4D*). Also, for most of the recorded neurons, reward responses emerged as the inactivation progressed (right panels of *Figure 4B, C and D*). In SR trials, firing rate to CS was unaltered by the injection. During the SC inactivation, DA responses evoked by the CS were not significantly different between LR and SR trials (*Figure 4—figure supplement 2*). Together, these results confirm that, in the absence of V1, visual signals signifying CS onset, with the capacity to elicit a short latency phasic response in presumed DA neurons, are most likely to be relayed via the direct retino-tecto-nigral projection (*Comoli et al., 2003*; *Dommett et al., 2005*), although an indirect contribution, possibly involving the pedunculopontine nucleus cannot be ruled out at present (*Harting, 1977*; *Redgrave et al., 1987*); *Kobayashi and Okada, 2007*).

## Discussion

In the present study, we investigated whether subcortical visual systems, in particular the midbrain superior colliculus (SC), can support behavioural Pavlovian conditioning, while at the same time evoke short latency phasic responses in ventral midbrain DA neurons. This was achieved by using monkeys with unilateral damage to the V1 that had been used previously to investigate the phenomenon of 'blindsight'. The purpose of using this preparation was to isolate the contribution of the SC that remained intact on the V1 disabled side. The main findings of the present study were, first, that after several days of training, presentation of a CS was equally capable of eliciting a robust conditioned response when it was presented either to the V1 lesion-affected visual field, or to the field served both by an intact visual cortex and the SC. This result demonstrated the capacity of residual subcortical visual pathways to elicit Pavlovian conditioned responses. Secondly, when identical CSs that predicted different amounts of primary reward (juice) were presented at different locations, either within the intact or lesion-affected visual fields, differential conditioned responses were elicited. This suggests that the subcortical neural mechanisms responsible for mediating the conditioned responses can discriminate CSs on the basis of spatial location. Thirdly, a critical involvement of the SC was established by showing that anticipatory conditioned responding reflecting reward expectation was disrupted when the critical locus representing the LR-CS within the spatial retinotopic map in the SC was locally inactivated with muscimol. Fourthly, parallel electrophysiological recording from putative DA neurons revealed that visual CSs presented to the lesion-affected visual field elicited patterns of phasic responses that have been widely reported by others. Specifically, the initial task-relevant fixation point evoked robust DA responses that were independent of subsequent CS value (*Bromberg-Martin et al., 2010*; *Matsumoto and Takada, 2013*); the temporally unpredicted CSs evoked phasic DA responses that were dependent on the predictive value of the CS (*Tobler et al., 2005*; *Fiorillo, 2013*); while the predicted reward deliveries evoked only muted responses (*Schultz, 1998*). Finally, phasic DA responses evoked by CS were almost completely abolished when the CS representation in the colliculus was pharmacologically blocked. Thus, the SC was critically involved in the short-latency activation of DA neurons by visual CSs presented to the V1 lesion-affected visual field. Together these results show that visual cues presented to the lesion-affected field in monkeys with a unilateral V1 lesion can support behavioral Pavlovian conditioning, and elicit DA responses that reflect the reward predicted by the CS via an afferent projection route involving the midbrain SC.

### Possible input pathways for reward prediction

Many studies have indicated that midbrain DA neurons causally contribute to reinforcement learning. For example, when reward expectation signals from DA neurons were impaired by D1 receptor blocker or when NMDA receptors were knocked out in DA receptor expressing neurons in various brain areas, conditioned response was impaired in many kinds of behavioral learning tasks (*Di Ciano et al., 2001*; *Flagel et al., 2011*; *Parker et al., 2010, 2011*; *Puig and Miller, 2012*; *Berridge and Robinson, 1998*). Alternatively, when DA neurons or neurons expressing D1 receptors were

activated by electrical or optogenetical stimulation, various forms of conditioned behaviour were induced (*Olds and Milner, 1954*; *Adamantidis et al., 2011*; *Ilango et al., 2014*; *Steinberg et al., 2013*; *Kravitz et al., 2012*). Thus, such involvement of dopaminergic transmission or DA neuron activity in learning has been well studied, however, it remains unclear how DA neurons are able to signal the value or salience of unpredicted objects or events at short-latency.

It has been proposed that the early phasic responses of DA neurons have two separable components; an early non-selective sensory response that represents temporal salient-event prediction errors, and a second component that codes the object/event's reward value (*Joshua et al., 2009*; *Bromberg-Martin et al., 2010*; *Schultz, 2016*). This view immediately provokes the question of what early afferent visual processing could allow the DA neurons to respond in this fashion to conditioned visual stimuli (the sensory modality that is most frequently used)? Following the onset of a visual CS response latencies in V1 are typically in the range 40–60 ms, while in the inferotemporal cortex where objects/events are identified they are slower in the range 80–100 ms (*Thorpe and Fabre-Thorpe, 2001*). Moreover, since there are no obvious direct connections to the ventral midbrain, the results of cortical visual processing are likely to be relayed via additional time consuming indirect routes. On the other hand, response latencies in the retino-recipient midbrain SC are significantly less (40–50 ms) and there is a direct tectonigral projection to substantia nigra pars compacta (*Comoli et al., 2003*; *McHaffie et al., 2006*; *May et al., 2009*). It is probable, therefore, that the earliest sensory component of the phasic DA response (70–150 ms) is mediated via subcortical visual processing involving the SC (*Comoli et al., 2003*; *Dommett et al., 2005*).

Two versions of the two-visual system hypothesis as an explanation for the bimodal characteristic of short latency phasic DA responses to visual CSs have been presented (Joshua et al., 2009; *Bromberg-Martin et al., 2010*; *Schultz, 2016*; *Redgrave et al., 2017*). The first is that the initial component of the phasic DA response is a non-selective salience signal that represents a temporal salient-event prediction error (Joshua et al 2009, *Bromberg-Martin et al., 2010*; *Schultz, 2016*). The second phasic component is value-coded and takes longer to compute because the unexpected event needs to be identified before its value is known. Stimulus identification frequently requires stimulus detection, foveation and cortical analysis of geometric form, colour, texture, and apparent motion, in various permutations and combinations (*Nomoto et al., 2010*). However, in the case of simple stimuli (for example, luminance change at different spatial locations) it is suggested that the non-selective salience and value components can merge to a near unimodal response that, in some cases, can be separated by sophisticated mathematical analysis (*Fiorillo et al., 2013*). This version suggests that for both subcortical salience and cortical stimulus identification the early sensory responses have to be relayed through an unspecified 'value-decoder' that communicates with DA neurons, thereby enabling them to report reward prediction errors (*Schultz, 2016*).

What is the likely location of the hypothesized 'value decoder'? Uchida and colleagues recently identified all the brain regions which project to DA neurons in rodents. They report afferent connections from the striatum, amygdala, subthalamic nucleus, pedunculopontine nucleus, rostromedial reticular nucleus, and GABAergic neurons in the substantia nigra pars reticulata (*Watabe-Uchida et al., 2012*). Consequently, there are many possible locations that receive input from primary visual structures, compute stimulus value and communicate this to DA neurons in the ventral midbrain. These indirect routes of communication can offer a perfectly reasonable explanation for the value coding of the second delayed component of the early phasic DA response.

However, it is important to note that the earliest component (70–150 ms) of phasic DA response is not always best described as a value insensitive salience signal. Both the present results (where cortical visual processing is impaired), and earlier studies of Schultz and his colleagues involving intact monkeys (*Tobler et al., 2005*; *Fiorillo, 2013*) report that when CSs can be discriminated on the basis of luminance change at different locations (a subcortical collicular visual competence – *Boehnke and Munoz, 2008*), the phasic DA response latencies are frequently around 100 ms (pre-gaze shift), unimodal and clearly code the predictive value of the CS. So how is it possible for unimodal phasic DA responses (for example, *Figure 1B – Tobler et al., 2005*) to code value at such short latencies? Visual response latencies in intermediary structures identified above are too long (typically >100 ms) to account for value coding of a unimodal phasic DA response that peaks at about 100 ms. A second, rather simpler version of two-visual system hypothesis can explain value-coding of both components of the early phasic DA response (*Redgrave et al., 2017*). The proposal is that the predictive value of a visual CS may already be encoded in the early sensory response of both the

cortical and subcortical early visual systems. For example, there are many papers that demonstrate that an association with, or an expectation of reward can dramatically influence the magnitude of the initial sensory response in early sensory areas throughout the brain (*Mogami and Tanaka, 2006*; *Serences and Saproo, 2010*; *Metzger et al., 2006*; *Leathers and Olson, 2012*), including the SC (*Ikeda and Hikosaka, 2003*). The most parsimonious explanation of how the earliest responses of DA neurons can be value-coded is, therefore, that they receive input from the SC that has been already value-coded through a classically conditioned process of sensory pre-tuning of the CS value in early sensory structures (*Ikeda and Hikosaka, 2003*).

Thus, in our study and those of others, stimuli are conditioned by Pavlovian association with different levels/probabilities of reward, prior to the recording of DA neurons (*Fiorillo et al., 2003*; *Tobler et al., 2005*; *Matsumoto and Hikosaka, 2009*). The likely effect of this process would be to tune the initial sensory responses in early visual structures to reflect the reward predicted by the CS. According to this suggestion, if the object/event prediction error detected in early visual structures has been value-coded by prior Pavlovian association, the event prediction error would also be a reward prediction error. In the case of the SC, if a value-coded signal evoked by a CS was relayed to the DA neurons via the tectonigral projection (*Comoli et al., 2003*; *Dommett et al., 2005*; *McHaffie et al., 2006*; *May et al., 2009*), it would explain how DA neurons can signal reward prediction errors with latencies in the range 70–150 ms (present study and *Fiorillo et al., 2003*; *Tobler et al., 2005*). On the other hand, in the case of complex CSs that are presented at the same location, or randomly at different locations, the SC would certainly detect the luminance change associated with CS onset, (*Boehnke and Munoz, 2008*). However, because subcortical sensory processing cannot perform complex CS discriminations (*Boehnke and Munoz, 2008*), this onset response will not be value-coded, which might explain why, with complex CSs, the initial sensory component of the DA phasic response is a non-selective salient-event prediction error. A possible explanation of the second value-coded component of the phasic DA response could be that the cortical processing responsible for object/event identification is equally subject to Pavlovian pre-tuning (*Mogami and Tanaka, 2006*; *Serences and Saproo, 2010*; *Weil et al., 2010*).

It is well known that there are two kinds of DA responses; one is sensitive to the value of future events, and the other is sensitive to their salience (*Matsumoto and Hikosaka, 2008*; *Lerner et al., 2015*; *Menegas et al., 2017*). In the context of the present study, we are unable to tell whether our DA responses reflected value or salience, because we used only reward associated CSs. To confirm which kinds of DA responses are elicited thorogh the subcortical visual processing, we have to conduct another experiments using aversive stimuli. However, at least, we could demonstrate that DA neurons could differentiate either reward value or salience with the visual information mediated by the SC.

## Materials and methods

### Subjects

Three adult Japanese monkeys (Macaca fuscata; all female, body weight 5–7 kg, monkey K, U and T) were used in this study. Details of the procedures for training and surgery of the monkeys have been described in previous reports (*Yoshida et al., 2008*; *Kato et al., 2011*). Briefly, under isoflurane anesthesia (1.0–1.5%), the monkeys were implanted with a holder with which the head was stabilized during the behavioural and electrophysiological experiments. The monkeys were allowed to recover for more than two weeks after surgery before pre-lesion training. All the experimental procedures were performed in accordance with the National Institutes of Health Guidelines for the Care and Use of Laboratory Animals and approved by the Committee for Animal Experiment at the National Institute of Natural Sciences.

### Unilateral V1 lesion

The right V1 of monkey K and U, and left V1 of monkey T were surgically removed by aspiration under isoflurane anesthesia (1.0–1.5%) (see *Yoshida et al., 2008*). The surgical operation was conducted before 46 months (monkey K), 44 months (monkey U), and six months (monkey T) from days when their training in this study was started. The opercular surface of the striate cortex and medial area in the Calcarine Sulcus was removed, while the ventrolateral part of the opercular surface, which

encodes foveal vision (visual field for eccentricity 0 to 1.0°) remained intact (*Figure 1A*, *Figure 1–figure supplement 1A and B*).

## Visually guided saccadic eye movement task

Prior to the surgery, animals were trained on a visually guided saccadic eye movement task. Their ability to respond to visual stimuli was assessed both before and after the V1 lesion. A monitor (Diamondcrysta WIDE RDT272WX (BK), MITSUBISHI) was positioned 34.5 cm in front of the monkeys' face. A real-time experimental control system (Tempo for Windows, Reflective Computing; http://reflectivecomputing.com/) was used for stimulus presentation and data collection. In this task, fixation point (FP) initially appeared at the center of monitor screen. Monkeys were required to maintain fixation in a window centered on the FP (size, 2.5° radius) for 1.6–2.0 s. A second target visual stimulus (0.6°) was then presented randomly at one of five possible locations in the hemi-visual field for two monkeys (monkey U and T) and one of three possible locations in visual hemifield for one monkey (monkey K) (*Figure 1—figure supplement 1C*). When the target appeared, the FP was extinguished and monkeys were required to make a saccade to the peripheral visual target. A window surrounding the target was a circle with a radius of half the distance between each target location (radius = eccentricity × sin (direction angle between neighboring target positions) / 2). This arrangement prevented the targets to overlap with each other. Target luminance Michelson contrast was 0.87–0.94 (13.4–31.3 Weber contrast) on a background of 1.0 cd / m$^2$. Reward was delivered if monkeys made a correct saccade to the target within 1 s after target presentation and maintaining fixation within the target window (3.2° radius) for 600 ms. Eye movements were measured with a video-based eye tracker (EYE-TRAC 6; Applied Science Laboratories, sampling rate: 240 Hz). All statistical analysis in this study were performed on Matlab (RRID:SCR_001622).

## Post-lesion assessment of visually-guided saccades

Details of the methods for calculations to construct the deficit map in these animals have been described previously (*Yoshida et al., 2008*). Luminance contrast of the targets was varied randomly trial-by-trial (0.02 to 0.9 as expressed in Michelson contrast (Weber contrast 0.04–18.0)). For this test, saccades landing in an area within a circle with a radius of half the distance between each target location (radius = eccentricity × sin(direction angle between neighboring target positions/2); 15° for monkey U and T, and 22.5° for monkey K) were counted as correct responses. The sensitivity of luminance contrast was defined as that representing the percentage of correct responses corresponding to the sensitivity value d' = 2 (threshold for luminance contrast) and deficit maps of individual monkeys were constructed with these values (*Figure 1—figure supplement 1C*). In general, the visual field disrupted by the lesion site extended from eccentricities about 5–20° in the monkeys used in this study. The luminance contrast and CS size were retained from previous studies that investigated visual responses of V1 neurons to stimuli presented in the natural blind spot. Our previous study also precluded the possibility of stray-light affecting the results in the present experimental environment by demonstrating the absence of a saccadic response to visual stimuli presented in the natural blind spot. The present Pavlovian conditioning experiments were initiated 46, 44 and 6 months after the V1 lesions in monkey K, U and T, respectively.

## Pavlovian conditioning task

The task sequence of the Pavlovian conditioning paradigm used in the present study is illustrated in *Figure 1b*. Conditioned stimuli (CS) (2.2° red square, luminance contrast: Michelson contrast 0.87 (Weber contrast 13.4) against the background of 1.0 cd / m$^2$) were presented in either the upper (eccentricity: 10°, direction: 45° relative to the horizontal axis from the FP) or lower quadrant (eccentricity: 10°, direction: −45° relative to the horizontal axis from the central FP) of the lesion-affected or intact visual hemifield. Experiments involving CS presentation to either the lesion-affected or intact visual hemifield were conducted on separate days. At the beginning of each trial, a fixation point (FP) appeared at the center of monitor. After a 0.7 to 1.2 s fixation period, a CS predicting a large reward (LR-CS) or a CS predicting a small reward (SR-CS) was presented for 1.0 or 1.7 s. The two CSs were pseudo-randomly alternated within a daily session. Throughout the task, monkeys were required to maintain their gaze on the central FP to assure that CS presentation was either to the lesion-affected, or intact visual hemi-field. If fixation was broken, the trial was terminated

immediately. The conditioned response (CR) in this task was the anticipatory licking elicited by the CS presentation that occurred prior to the juice delivery. The CR was measured by detecting electric contact between the monkey and the reward tube or by a photo-detector in experiments involving electrophysiological recording. A lick was recorded when the monkeys' tongue was observed to approach the reward spout. To quantify the conditioned response elicited by the visual CS, the number of licking responses detected during the cue presentation (0 to 1.3 s) was counted in 0.1 s time bins in 14–16 sessions for each hemifield of each monkey. The frequency of licking (licking rate) was compared to a baseline frequency during the 1 s period (−1 to 0 s) before the CS onset (one-tailed paired t test, significant level at p<0.05).

## Recording from DA neurons

A principal aim of the study was to record from single DA neurons while the monkeys were engaged in the Pavlovian conditioning task. This was achieved using epoxylite-coated tungsten microelectrode (impedance: 9–10 MΩ at 1 kHz, FHC). Voltage recording were bandpass-filtered between 0.1 (or 0.3) and 10 kHz. Standard criteria were used for identification of putative DA neurons (*Ungless et al., 2004*). First, the location of SNc and the VTA were estimated from MR images taken in advance. After having isolated a single neuron in the appropriate region, we tested whether the presentation of an unpredicted reward would cause a response. Two criteria to confirm the likelihood that we were recording from a DA neuron; (1) it had a low baseline activity between 1.0–10.0 Hz (*Schultz and Romo, 1987*; *Matsumoto and Hikosaka, 2009*); and (2) the neuron had a spike width, which was clearly longer than those of nearby neurons in the substantia nigra pars reticulata (SNr) that had rates of baseline firing in excess of 40 Hz (*Ungless et al., 2004*; *Matsumoto and Takada, 2013*).

## Muscimol injections

To determine the role of the residual subcortical visual circuit in eliciting conditioned responses in the Pavlovian task and CS-evoked responses in DA neurons we conducted experiments in which the SC on the V1 lesion-affected side was inactivated. In a previous study with these subjects (*Kato et al., 2011*) reported that the monkeys were unable to make saccades to parts of the visual field injected locally with the gamma aminobutyric acid A (GABA$_A$) receptor agonist, muscimol. In our experiments we used additional single unit electrophysiology to locate the response field of the SC neurons responsive to the LR-CS. At these sites muscimol (0.5 µg in 0.5 µL) was pressure-injected (0.4 µL/min) using a 10 µL Hamilton syringe (Hamilton Company, Reno, Nevada, USA) mounted in a syringe pump. Conditioned response was measured both before and during inactivation of the SC.

In some experiments we recorded the activity of presumed DA neurons while the animals were performing the Pavlovian task. Then, the SC was injected with muscimol. After recording DA activity for about 60 CS presentations, muscimol was injected into the SC while recording from the same neuron was maintained. In some sessions, post-injection trials started immediately after the injection, while in others they started 10 to 20 min after the injection.

## Histology

After all behavioural testing and electrophysiological recording had been completed with monkey K, two small electrolytic lesions were made in each recording track (20 mA, 30 s). The animal was then euthanized and coronal sections (40 µm) of tissue that included SNc were immunostained for tyrosine hydroxylase (TH) to reveal the location of DA neurons (*Figure 3D*). (RRID:AB_390204 for the antibody)

## Acknowledgements

We thank M Togawa, Y Yamanishi, N Takahashi, T Kuwahara, and K Isa for technical assistance.

# Additional information

## Funding

| Funder | Grant reference number | Author |
| --- | --- | --- |
| Japan Society for the Promotion of Science | KAKENHI Grant Number 22220006 | Tadashi Isa |
| Ministry of Education, Culture, Sports, Science, and Technology | 26112008 | Tadashi Isa |
| Japan Society for the Promotion of Science | KAKENHI Grant Number 26221003 | Tadashi Isa |
| Japan Agency for Medical Research and Development | Strategic Research Program for Brain Sciences 100160600067 | Tadashi Isa |

The funders had no role in study design, data collection and interpretation, or the decision to submit the work for publication.

## Author contributions

NT, Conceptualization, Data curation, Formal analysis, Investigation, Methodology, Writing—original draft, Writing—review and editing; RK, Conceptualization, Investigation, Methodology, Writing—review and editing; PR, Conceptualization, Methodology, Project administration, Writing—review and editing; TI, Conceptualization, Funding acquisition, Investigation, Methodology, Project administration, Writing—review and editing

## Author ORCIDs

Norihiro Takakuwa, http://orcid.org/0000-0003-4075-5697
Tadashi Isa, http://orcid.org/0000-0001-5652-4688

## Ethics

Animal experimentation: All the experimental procedures were performed in accordance with the National Institutes of Health Guidelines for the Care and Use of Laboratory Animals and approved by the Committee for Animal Experiment at the National Institute of Natural Sciences (Permit Number: 16A060).

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
