## [Decision Letter]

Thank you for submitting your article "Emergence of value coded dopamine signals via the superior colliculus in blindsight monkeys" for consideration by *eLife*. Your article has been favorably evaluated by David Van Essen (Senior Editor) and three reviewers, one of whom, Naoshige Uchida (Reviewer #1), is a member of our Board of Reviewing Editors. The following individual involved in review of your submission has agreed to reveal their identity: Ethan Bromberg Martin (Reviewer #3).

The reviewers have discussed the reviews with one another and the Reviewing Editor has drafted this decision to help you prepare a revised submission.

Summary:

The mechanisms underlying the generation of dopamine responses remain to be clarified. This study examined the role of the superior colliculus (SC) in the generations of conditioned responses (CRs) (licking) as well as of the dopamine response evoked by conditioned stimuli (CS) in monkeys with V1 lesions (a model of blindsight). The authors first show that (1) a reward-predictive CS elicited a similar level of CRs (licking) regardless of whether the stimulus was presented in the affected or unaffected visual field. (2) Furthermore, CSs predicting a larger amount of reward (LR) elicited stronger licking behavior compared to that predicting small reward (SR). The authors then recorded the activity of putative dopamine neurons in the substantia nigra pars compacta (SNc), and showed that (3) dopamine neurons exhibited phasic responses to the reward predictive CS presented in the affected visual field, which depended on the size of predicted reward. Finally, the authors demonstrate that (4) inactivation of SC greatly reduced the dopamine responses to reward predictive CSs but not those to the presentation of a fixation point or reward.

All the reviewers thought that this study addresses an important question, and results are generally convincing. However, they found a number of interpretational issues, and had different levels of enthusiasm with regard to the depths of the study and the generality of the results due to the use of blindsight model monkeys. Nevertheless, all agreed that the data contain important information. The following essential points, however, need to be addressed by revising the text and by analyzing the existing data for this work to be published in *eLife*.

Essential points:

1) Justification of blindsight should be done more carefully. That is, the effects of V1 lesions on behavior need to be explained in a greater detail. In Figure 1—figure supplement 1, the authors show some data measuring the visual sensitivity. The authors report the procedures in Materials and methods, and do not include these materials in the main text. It would be helpful if the authors summarize these results in the Results section in some depths and justify blindsights based on the data (as well as previous studies). Furthermore, please summarize the phenotypes of blind-sight model monkeys studied previously in Introduction, so that the readers can understand the nature of the monkey model of blindsight.

2) It is unclear whether and what effects V1 lesions have had on dopamine responses. Have the authors recorded the activity of dopamine neurons while presenting the CSs in the unaffected visual field? If so, it would be very helpful to compare the dopamine responses to CSs presented in the affected and unaffected visual field. How do the responses compare when stimuli are presented to the two hemispheres?

3) The results demonstrated the necessity of the SC in conditioned responses and dopamine responses, but did not prove that its involvement was via direct retino-tecto-nigral projection to dopamine neurons. Also, the results show that SC activity is necessary to construct a value signal in this task, but does not prove that SC activity itself signaled value or sent that signal to dopamine neurons. The authors should more explicitly discuss these alternative possibilities. For further details, please look at reviewer 3's comment #1.

4) Figure 4 only shows the responses to CS/outcome for LR. But it is also important to show the responses to CS/outcome for SR because this will provide crucial information regarding whether discrimination between the two CSs is impaired, not just the response to any single CS.

5) The authors refer to the cue-evoked responses as 'value-coded' responses. Strictly speaking, however, the experimental design does not allow the authors to distinguish responses due to value versus salience, with salience defined by the absolute, unsigned value. To distinguish these possibilities, one has to test whether a cue that predicts an aversive event excites or inhibits a given neuron. The authors should discuss both of these possibilities. This issue is important because some evidence suggests that dopamine neurons in the lateral SNc or those projecting to the posterior striatum may predominantly signal salience (Matsumoto and Hikosaka, 2008; Lerner et al., 2016; Menegas et al., 2017).

On a related note, Figure 3 indicates the latency of the phasic response, as defined by an increase in the impulse rate by an amount greater than 2SD. This measure doesn't necessarily reflect the latency to value coding. Recent studies have highlighted the fact that phasic DA response is comprised of an early, non-value coding component that may reflect physical salience and novelty, and a later component that codes value (Fiorillo et al., 2013; Lak et al., 2016; Matsumoto and Hikosaka, 2009; Nomoto et al., 2010). It would be good to acknowledge this distinction, as it could be important to understand what information exactly is coded by the circuit. Moreover, for value coding, a better measure would be to determine the latency to differential responses. What is the earliest time point that authors see value differentiation? Does the V1 or SC lesion affect this latency?

6) Subsection “Possible input pathways for reward expectation”, last paragraph. Please do not use the term "sensory pre-conditioning', since it refers to a specific kind of training without reinforcement (Brogden, 1939), which isn't what the authors want to say here.

7) There are many places where citations are inappropriate. Some were listed in the referees' comments. Please correct citations throughout the manuscript.

8) Time course of muscimol effects. As clearly stated in the text and clearly observable in Figure 4, the muscimol effects developed over time. Can the authors separate the slow onset of muscimol inactivation from a learning effect? Another possibility is that muscimol inactivation disrupted the perceived contingency between the CS and US, which in turn induced extinction. Do the authors know the time course of muscimol inactivation in the SC so that the time courses of the effect on SC neurons and dopamine neurons be directly compared?

Reviewer #1:

This study examined the role of the superior colliculus (SC) in the generation of dopamine responses as well as monkeys' behavioral response to reward-predictive cues. The authors used monkeys with V1 lesions, blind-sight model monkeys, to examine this, instead of normal monkeys. The authors first show that (1) blind-sighted monkeys can still exhibit relatively normal anticipatory licking behavior in response to reward-predictive cues after training, that (2) dopamine neurons in the substantia nigra pars compacta (SNc) are excited by reward-predictive cues. The authors also demonstrate that (3) inactivation of SC reduced the conditioned response (anticipatory licking) as well as (4) dopamine neurons' response to reward-predictive cues.

The neural mechanism that generates dopamine responses during behavior remains to be clarified, and this study addresses this important question. I have some interpretational issues (see below). Furthermore, the use of blind-sight monkeys somewhat restricts the impact of this study. Nonetheless, the results are overall convincing. I believe that after revising the text, this study will be appropriate for publication in *eLife*.

1) The authors refer to the cue-evoked responses as 'value-coded' responses. Strictly speaking, however, the experimental design does not allow the authors to distinguish responses due to value versus salience, with salience defined by the absolute, unsigned value. To distinguish these possibilities, one has to test whether a cue that predicts an aversive event excites or inhibits a given neuron. I suggest that the authors discuss both of these possibilities. This issue is important because increasing evidence suggests that dopamine neurons in the lateral SNc may predominantly signal salience (Matsumoto and Hikosaka, 2008; Lerner et al., 2016; Menegas et al., 2017).

Reviewer #2:

The current manuscript presents a fundamentally interesting and novel approach to study reward processing. The authors use a monkey model of 'blind-sight' wherein the authors lesion V1, and then monkeys perform a reward prediction task. Visual cues were used as reward predictors (predicting large and small rewards) During behavioral measurement and neuronal recording, muscimol was injected into the SC. Muscimol inactivation affected behavioral and neuronal correlates of reward processing.

If this summary seems vague, it is because I have trouble understanding the implications of doing both V1 lesions and muscimol inactivation. Accordingly, despite my strong conviction that this study is fundamentally interesting, I have equally strong reservations about the methods and the results.

1) My first major and overarching concern with this study is this: What was the effect of the V1 lesion as regards the results presented here? It is clear throughout the manuscript that muscimol injection into the SC affects dopamine reward processing, but it is not at all clear that the V1 lesion played any role. The lick response and saccade behavior seem normal (Figure 1, Figure 2). Figure 1—figure supplement 1 notes reduced contrast sensitivity, but the implications of this for reward processing are not clear. Am I missing something?

2) It appears that during dopamine recordings, the stimuli were only presented on the V1 lesion affected side, is this true? The entirety of Figure 3 seems to demonstrate that dopamine neuron responses reflect the larger and smaller value. This is such a well-known result, so I wonder why an entire figure is devoted to it? How do the responses compare when stimuli are presented to the two hemispheres? The entirety of Figure 4 appears to show that muscimol injections into the SC abolish a large fraction of dopamine signaling. Would this also be true for reward-predicting stimuli presented to the intact hemisphere? Again, what is the role of the V1 lesion in this study?

3) The dopamine response time course. Figure 3 indicates the latency of the phasic response, as defined by an increase in the impulse rate by an amount greater than 2SD. This measure doesn't necessarily reflect the latency to value coding. Recent studies have highlighted the fact that phasic DA response is comprised of an early, non-value coding component that may reflect physical salience and novelty, and a later component that codes value (Fiorillo et al., 2013; Lak et al., 2016; Matsumoto and Hikosaka, 2009; Nomoto et al., 2010). It would be good to acknowledge this distinction, as it could be important to understand what information exactly is coded by the circuit. Moreover, for value coding, a better measure would be to determine the latency to differential responses. What is the earliest time point that authors see value differentiation? Does the V1 lesion affect this latency?

4) Time course of muscimol effects. As clearly stated in the text and clearly observable in Figure 4, the muscimol effects developed over time. Can you separate the slow onset of muscimol inactivation from a learning effect? Said in another way, could muscimol inactivation disrupt the perceived contingency between the CS and US, such that extinction occurs? Would this not also explain the results in Figure 4? Halorhodopsin has been successfully applied to the NHP colliculus (Cavanaugh et al., 2012), why not use a modern technique that allows for precise manipulation?

Reviewer #3:

In this interesting paper, Takukawa and colleagues test several longstanding hypotheses about the routes through which sensory signals can be used to compute the "reward prediction error" signals that are encoded by dopamine neurons. They take advantage of their fascinating 'blindsight monkey' preparation to show several novel results: (1) animals can learn to predict rewards based on the location of a stimulus within the 'blind', V1-lesioned hemifield, as assessed by anticipatory licking, (2) DA neurons can do so as well, (3) both these reward-related behaviors and DA responses require an intact representation of the visual stimulus in the SC.

This is a valuable study of the pathways that control midbrain DA responses. Not only is the demonstration of intact DA signals despite V1 lesion already valuable on its own, but the authors also went the extra mile to test the causal contribution of the SC in this process. This study is also valuable due to its relevance to the short-latency response component of DA neurons, which many studies tend to gloss over. The possible implication for reward processing in the absence of phenomenal awareness is also intriguing – I wonder if it would be possible to do the behavioral experiment in blindsighted humans?

Overall, the motivation, experiments, and conclusions seem generally solid. I do have a few concerns about the interpretations that I suggest dialing back slightly, and a few suggestions about the analysis and presentation.

1) "These results indicate that the retino-tecto-nigral projections can relay reward-predicting visual information to DA neurons and integrity of the SC is necessary for visually-elicited classically conditioned responses after V1 lesion." "Specifically, our results show that visual input to DA neurons from the SC can signal value coded sensory events."

These and related statements in the paper seem a bit too strong.

The authors showed a clear necessity for the SC, but did not prove that its involvement was via direct retino-tecto-nigral projection to DA neurons. It could be an indirect effect. The authors argue against an indirect effect in the Discussion, but as they note, this is not conclusive proof.

Also, this paper proves that SC activity is necessary to construct a value signal in this task, but does not prove that SC activity itself signaled value or sent that signal to DA neurons. For instance, the SC's role in contributing to DA signals could be to code the stimulus location, which is the value-relevant feature in this task. A different area would then read out that location information and use it to figure out the stimulus value.

At least some of the SC's contribution in this task seems likely to occur in this manner. The SC is well known to be important for visual attention. Suppose SC activity is required for the brain to properly process a visual stimulus. Inactivating the SC would interfere with licking behavior and value signals in DA neurons, but that could reflect a more general deficit, not value coding by the SC.

Finally, let's suppose that the SC does directly transmit value coded signals to DA neurons. I feel that the way the authors explain this concept may mislead some readers. The authors describe it in terms of the retino-tecto-nigral pathway, and with an emphasis on Pavlovian training. This makes it sound like the value signals emerge via slow, long-term plasticity at synapses in this pathway (retina-SC or SC-DA):

"There are many papers that demonstrate that an association with, or an expectation of reward can dramatically influence the magnitude of the initial sensory response in primary sensory areas throughout the brain (Mogami and Tanaka, 2006; Serences and Saproo, 2010; Metzger et al., 2006; Leathers and Olson, 2012), including the SC (Ikeda et al., 2003). […] The magnitude of such responses would then reflect the value of the predicted reward."

However, in my view value-related signals in the SC are more likely to emerge due to short-term changes in SC response gain caused by top-down modulatory inputs from other areas (e.g. spatial attention based on reward value). This seems more compatible with the Ikeda study cited here, since the rewarded location changed rapidly during that experiment, which the SC can adapt to in < 3 trials (Isoda and Hikosaka, Journal of Neurophysiology 2008). If so, then despite the authors' apparent emphasis on bottom-up retino-tecto-nigral transmission, the value signals are actually injected into this pathway by other (possibly cortical) brain areas that send a top-down modulatory signal to the SC to change its response gains at different parts of its visual map.

P.S. I suggest not using the term "sensory pre-conditioning", since it refers to a specific kind of training without reinforcement (Brogden, 1939), which isn't what the authors want to say here.

2) Figure 4 shows the responses to CS/outcome for LR, which look beautiful. But it is also important to show the responses to CS/outcome for SR, even though only the LR CS was presented in the inactivated part of the visual field. This is because the key point is whether discrimination between the two CSs is impaired, not just the response to any single CS. Indeed, this distinction was important in the analysis of behavior: while inactivation abolished one animal's licking to the LR CS, for the other animal it induced equal licking for both CSs. Also, if the authors do see a change in DA responses to the SR CS/outcome despite it being outside the inactivated field, that would be interesting!

[Editors' note: further revisions were requested prior to acceptance, as described below.]

Thank you for resubmitting your work entitled "Emergence of value coded dopamine signals via the superior colliculus in V1 lesioned monkeys" for further consideration at *eLife*. Your revised article has been favorably evaluated by David Van Essen (Senior Editor), a Reviewing Editor, and one reviewer.

The manuscript has been improved but there are some remaining issues that need to be addressed before acceptance, as outlined below:

Essential points:

1) The authors still emphasize subjective awareness but this is not very well documented or backed up by data. It is also noted that the lesions might have been made a long time ago (~10 years?). Given these problems, we would like the authors to further address these points. Do the monkeys recover any (conscious) visual abilities (It wouldn't need to be high acuity vision to complete this task)? Was the data in Figure 1—figure supplement 1 about contrast sensitivity recorded recently, or is it adapted from the 2008 paper? In addition to discussing possible recovery, the authors need to be much clearer (throughout the entire paper) about the limitations of this study in the absence of psychophysical tests.

2) The same problem, regarding phenomenal awareness, is also important for the data in Figure 4 (right, reward response). This data demonstrate that the SC is necessary for visual reward prediction, but it appears impossible to know whether the muscimol inactivation of SC caused loss of phenomenal awareness, or whether the muscimol inactivation was done in the absence of phenomenal awareness and thus constitutes a neural mechanism for unconscious reward processing. The authors suggest the latter (end of subsection “Possible input pathways for reward prediction”), but it does not seem to be an appropriate conclusion (or suggestion) because the authors did not measure conscious awareness.

3) Although the authors acknowledge that their paradigm cannot discriminate between salience and value, the title of the paper still reads as value-coding. Because the cues probably had very similar physical properties, and hence similar physical salience, this problem can be dealt with by more careful wording throughout. It appears to be confusing if the authors just add a disclaimer paragraph to the Discussion, and ignoring the disclaimer everywhere else. We suggest that the authors remove "value-coded" and then use more careful wording and make the manuscript consistent throughout the manuscript.

---

## [Author Response]

Essential points:

1) Justification of blindsight should be done more carefully.

The purpose of our investigation was to test whether evolutionary archaic subcortical visual circuitry can support Pavlovian conditioning and can activate ventral midbrain dopamine neurons. To test this, it was necessary to have a preparation where the cortical visual system was disabled. This would permit us to ascribe residual visual competences to subcortical visual systems. The fact that this preparation in primates has also been associated with the phenomenon of ‘blindsight’ is not strictly relevant to the present study. The wording has been changed throughout the manuscript to make this point clear. That said, it was noted correctly by reviewer #3 that our results do also extend the range of residual visual competences that can be ascribed to V1 lesioned primates, and will therefore also be of interest to the ‘Blindsight’ research community. Consequently we have kept the description of blindsight (subsection “Possible input pathways for reward prediction”)

That is, the effects of V1 lesions on behavior need to be explained in a greater detail. In Figure 1—figure supplement 1, the authors show some data measuring the visual sensitivity.

“Blindsight” is not the main goal of this study. This was included simply to confirm the functional efficacy of the V1 lesion in much the same way as the results of structural MRI are included to confirm the lesion’s location.

The authors report the procedures in Materials and methods, and do not include these materials in the main text. It would be helpful if the authors summarize these results in the Results section in some depths and justify blindsights based on the data (as well as previous studies). Furthermore, please summarize the phenotypes of blind-sight model monkeys studied previously in Introduction, so that the readers can understand the nature of the monkey model of blindsight.

We have added information about phenotypes of an animal model of blindsight to introduction and results. The text has now been modified to make it clear the animal model was used for the disabling of cortical visual processing, rather than the behavioural phenomenon of ‘blindsight’. (Introduction, third paragraph; Results, first paragraph). We also changed the title, “Emergence of value coded dopamine signals via the superior colliculus in V1 lesioned monkeys”.

2) It is unclear whether and what effects V1 lesions have had on dopamine responses.

To establish the SC can relay visual information that can induce normal DA responses, we have compared the responses to CSs in intact (V1 + SC both functional) and in affected visual field (where only the SC is functional). We have added 2 figure panels: one is the earliest time points that value differentiation emerged between LR and SR trials, and the other compares firing rates between them.

Have the authors recorded the activity of dopamine neurons while presenting the CSs in the unaffected visual field? If so, it would be very helpful to compare the dopamine responses to CSs presented in the affected and unaffected visual field. How do the responses compare when stimuli are presented to the two hemispheres?

We have added information about DA responses to CSs presented into intact visual field. First, we have calculated the earliest time points that value differentiation between LR and SR trials can be detected (Figure 3). Secondly, the firing rate of CS-evoked responses were compared (Figure 3—figure supplement 1). Together these data show clearly that the V1 lesion did not strongly affect DA responses to visual CSs.

3) The results demonstrated the necessity of the SC in conditioned responses and dopamine responses, but did not prove that its involvement was via direct retino-tecto-nigral projection to dopamine neurons.

The text has been modified accordingly (Abstract; Introduction, fourth and last paragraphs; subsection “Recording from DA neurons”).

Also, the results show that SC activity is necessary to construct a value signal in this task, but does not prove that SC activity itself signaled value or sent that signal to dopamine neurons. The authors should more explicitly discuss these alternative possibilities. For further details, please look at reviewer 3's comment #1.

Response: A detailed description of these points has now been made in the Discussion section of the manuscript (subsection “Possible input pathways for reward prediction”).

4) Figure 4 only shows the responses to CS/outcome for LR. But it is also important to show the responses to CS/outcome for SR because this will provide crucial information regarding whether discrimination between the two CSs is impaired, not just the response to any single CS.

The responses to CS(SR) and SR have been calculated and now, the responses to both LR- and SR-CSs have been compared before and after muscimol injection (Figure 4—figure supplement 2). After muscimol injection, there was no significant difference between LR and SR trials. But because responses to SR-CS was not significantly different between before and after injection, we cannot determine whether CSs discrimination was impaired or not.

5) The authors refer to the cue-evoked responses as 'value-coded' responses. Strictly speaking, however, the experimental design does not allow the authors to distinguish responses due to value versus salience, with salience defined by the absolute, unsigned value. To distinguish these possibilities, one has to test whether a cue that predicts an aversive event excites or inhibits a given neuron. The authors should discuss both of these possibilities. This issue is important because some evidence suggests that dopamine neurons in the lateral SNc or those projecting to the posterior striatum may predominantly signal salience (Matsumoto and Hikosaka, 2008; Lerner et al., 2016; Menegas et al., 2017).

A paragraph on the difference between value and salience coding is included in the Discussion (subsection “Possible input pathways for reward prediction”, seventh paragraph). In the context of the present study, we are unable to tell whether our DA responses reflected value or salience, but we could demonstrate that DA neurons could differentiate either reward value or salience.

On a related note, Figure 3 indicates the latency of the phasic response, as defined by an increase in the impulse rate by an amount greater than 2SD. This measure doesn't necessarily reflect the latency to value coding. Recent studies have highlighted the fact that phasic DA response is comprised of an early, non-value coding component that may reflect physical salience and novelty, and a later component that codes value (Fiorillo et al., 2013; Lak et al., 2016; Matsumoto and Hikosaka, 2009; Nomoto et al., 2010). It would be good to acknowledge this distinction, as it could be important to understand what information exactly is coded by the circuit. Moreover, for value coding, a better measure would be to determine the latency to differential responses. What is the earliest time point that authors see value differentiation? Does the V1 or SC lesion affect this latency?

A new panel has been added to Figure 3) in which the requested data are presented.

6) Subsection “Possible input pathways for reward expectation”, last paragraph. Please do not use the term "sensory pre-conditioning', since it refers to a specific kind of training without reinforcement (Brogden, 1939), which isn't what the authors want to say here.

We removed the sentences including the term.

7) There are many places where citations are inappropriate. Some were listed in the referees' comments. Please correct citations throughout the manuscript.

The citations have been revised as requested by the reviewers.

8) Time course of muscimol effects. As clearly stated in the text and clearly observable in Figure 4, the muscimol effects developed over time. Can the authors separate the slow onset of muscimol inactivation from a learning effect? Another possibility is that muscimol inactivation disrupted the perceived contingency between the CS and US, which in turn induced extinction. Do the authors know the time course of muscimol inactivation in the SC so that the time courses of the effect on SC neurons and dopamine neurons be directly compared?

It is hard to separate time courses of muscimol inactivation in the SC and of learning effect. Because we did not record activity in the SC and assessment of the effect of muscimol was performed at the end of each experiment. Furthermore, it is not clear what the animal would learn with the progressive onset of inhibition by muscimol in the SC. In our experimental paradigm the CS consistently predicts an upcoming reward….this does not change throughout the experimental session. What does change is that one monkey failed completely to respond behaviourally to the CS while the other monkey was no longer able to discriminate the location of CS presentation. In the case of the dopamine neurons – both the individual and population averaged data show that when both the V1 and the SC were disabled there was no phasic response to CS onset. With both cortical and subcortical visual systems out of action this would be expected. Further evidence supporting this interpretation was the return of the phasic dopamine response to reward presentation, again in both individual and population data. In intact preparations, the response to reward is suppressed if a CS predicts it. The fact that in our experiments the response to reward returned after muscimol inactivation of the SC is indicative that the CS was not detected. Again we have tried to make this point clearer in the Discussion section.

Reviewer #1:

[…] The neural mechanism that generates dopamine responses during behavior remains to be clarified, and this study addresses this important question. I have some interpretational issues (see below). Furthermore, the use of blind-sight monkeys somewhat restricts the impact of this study. Nonetheless, the results are overall convincing. I believe that after revising the text, this study will be appropriate for publication in eLife.

1) The authors refer to the cue-evoked responses as 'value-coded' responses. Strictly speaking, however, the experimental design does not allow the authors to distinguish responses due to value versus salience, with salience defined by the absolute, unsigned value. To distinguish these possibilities, one has to test whether a cue that predicts an aversive event excites or inhibits a given neuron. I suggest that the authors discuss both of these possibilities. This issue is important because increasing evidence suggests that dopamine neurons in the lateral SNc may predominantly signal salience (Matsumoto and Hikosaka, 2008; Lerner et al., 2016; Menegas et al., 2017).

A large section has been added to the Discussion that speaks to this point directly (subsection “Possible input pathways for reward prediction”, seventh paragraph).

Reviewer #2:

The current manuscript presents a fundamentally interesting and novel approach to study reward processing. The authors use a monkey model of 'blind-sight' wherein the authors lesion V1, and then monkeys perform a reward prediction task. Visual cues were used as reward predictors (predicting large and small rewards) During behavioral measurement and neuronal recording, muscimol was injected into the SC. Muscimol inactivation affected behavioral and neuronal correlates of reward processing.

If this summary seems vague, it is because I have trouble understanding the implications of doing both V1 lesions and muscimol inactivation. Accordingly, despite my strong conviction that this study is fundamentally interesting, I have equally strong reservations about the methods and the results.

This comment is a further consequence of the misleading use of the term ‘blindsight’– apologies and hopefully now corrected.

1) My first major and overarching concern with this study is this: What was the effect of the V1 lesion as regards the results presented here? It is clear throughout the manuscript that muscimol injection into the SC affects dopamine reward processing, but it is not at all clear that the V1 lesion played any role. The lick response and saccade behavior seem normal (Figure 1, Figure 2). Figure 1—figure supplement 1 notes reduced contrast sensitivity, but the implications of this for reward processing are not clear. Am I missing something?

We are sorry, this is a recurrence of the misleading – our fault – that it was important to perform the experiments in ‘blindsight’ subjects rather than subjects in which the cortical visual system had been disabled. The title and the text have been changed accordingly.

2) It appears that during dopamine recordings, the stimuli were only presented on the V1 lesion affected side, is this true? The entirety of Figure 3 seems to demonstrate that dopamine neuron responses reflect the larger and smaller value. This is such a well-known result, so I wonder why an entire figure is devoted to it? How do the responses compare when stimuli are presented to the two hemispheres? The entirety of Figure 4 appears to show that muscimol injections into the SC abolish a large fraction of dopamine signaling. Would this also be true for reward-predicting stimuli presented to the intact hemisphere? Again, what is the role of the V1 lesion in this study?

We have added two figures in which DA responses to CSs in intact and lesion-affected visual field were compared (Figure 3 and Figure 3—figure supplement 1). But we did not conduct the experiments; muscimol injection into contralesional SC. This is interesting question that has been addressed in the future study.

3) The dopamine response time course. Figure 3 indicates the latency of the phasic response, as defined by an increase in the impulse rate by an amount greater than 2SD. This measure doesn't necessarily reflect the latency to value coding. Recent studies have highlighted the fact that phasic DA response is comprised of an early, non-value coding component that may reflect physical salience and novelty.

This is only when value cannot be discriminated by sub-cortical systems at short latency).

And a later component that codes value (Fiorillo et al., 2013; Lak et al., 2016; Matsumoto and Hikosaka, 2009; Nomoto et al., 2010).

This is only true when complex stimuli are used.

It would be good to acknowledge this distinction, as it could be important to understand what information exactly is coded by the circuit. Moreover, for value coding, a better measure would be to determine the latency to differential responses. What is the earliest time point that authors see value differentiation? Does the V1 lesion affect this latency?

A significant part of the amended Discussion addresses this point.

In addition, we have reanalyzed DA responses to the LR- and SR-CSs, and have revise Figure 3 accordingly. The new data include the time period when there is significant difference (>15ms) between response to LR-CS and SR-CS. However, the earliest time points that value differentiation became apparent were not much different when CS appeared in the lesion-affected or the intact visual field.

4) Time course of muscimol effects. As clearly stated in the text and clearly observable in Figure 4, the muscimol effects developed over time. Can you separate the slow onset of muscimol inactivation from a learning effect? Said in another way, could muscimol inactivation disrupt the perceived contingency between the CS and US, such that extinction occurs?

The suggestion here is that muscimol is disrupting the contingency relationship between CS and UCS but requires that both be perceived. Given that on the lesioned side there is no visual cortex and an inactivated SC, what would be the neural substrate that would be able to detect the presence of the visual CS? Again this point has been addressed in the amended Discussion (subsection “Possible input pathways for reward prediction”).

Reviewer #3:

[…] This is a valuable study of the pathways that control midbrain DA responses. Not only is the demonstration of intact DA signals despite V1 lesion already valuable on its own, but the authors also went the extra mile to test the causal contribution of the SC in this process. This study is also valuable due to its relevance to the short-latency response component of DA neurons, which many studies tend to gloss over. The possible implication for reward processing in the absence of phenomenal awareness is also intriguing – I wonder if it would be possible to do the behavioral experiment in blindsighted humans?

If possible, we would like to do this.

Overall, the motivation, experiments, and conclusions seem generally solid. I do have a few concerns about the interpretations that I suggest dialing back slightly, and a few suggestions about the analysis and presentation.

1) "These results indicate that the retino-tecto-nigral projections can relay reward-predicting visual information to DA neurons and integrity of the SC is necessary for visually-elicited classically conditioned responses after V1 lesion." "Specifically, our results show that visual input to DA neurons from the SC can signal value coded sensory events."

These and related statements in the paper seem a bit too strong.

We have responded to this comment with significant alterations to the Discussion section – i.e. we now present two interpretations of how sensory input from the SC could evoke value-coded responses in DA neurons.

The authors showed a clear necessity for the SC, but did not prove that its involvement was via direct retino-tecto-nigral projection to DA neurons. It could be an indirect effect. The authors argue against an indirect effect in the Discussion, but as they note, this is not conclusive proof.

We take this point and have amended the text accordingly (Abstract; Introduction, fourth and last paragraphs; subsection “Recording from DA neurons”).

Also, this paper proves that SC activity is necessary to construct a value signal in this task, but does not prove that SC activity itself signaled value or sent that signal to DA neurons. For instance, the SC's role in contributing to DA signals could be to code the stimulus location, which is the value-relevant feature in this task. A different area would then read out that location information and use it to figure out the stimulus value.

This interpretation is now included in the discussion (subsection “Possible input pathways for reward prediction”) as one of the possibilities for explaining value coding of DA responses.

At least some of the SC's contribution in this task seems likely to occur in this manner. The SC is well known to be important for visual attention.

Indeed it is, and that the direction of attention is powerfully influenced by stimulus value – gaze shifts are made to the most valuable stimuli, which suggests that when this occurs value must be known before, not after the gaze shift.

Suppose SC activity is required for the brain to properly process a visual stimulus. Inactivating the SC would interfere with licking behavior and value signals in DA neurons, but that could reflect a more general deficit, not value coding by the SC.

This point is asking questions about exactly how the SC contributes to triggering conditioned responses and phasic DA responses. This is a very important issue and is part of our future programme of work in this area.

Finally, let's suppose that the SC does directly transmit value coded signals to DA neurons. I feel that the way the authors explain this concept may mislead some readers. The authors describe it in terms of the retino-tecto-nigral pathway, and with an emphasis on Pavlovian training. This makes it sound like the value signals emerge via slow, long-term plasticity at synapses in this pathway (retina-SC or SC-DA):

"There are many papers that demonstrate that an association with, or an expectation of reward can dramatically influence the magnitude of the initial sensory response in primary sensory areas throughout the brain (Mogami and Tanaka, 2006; Serences and Saproo, 2010; Metzger et al., 2006; Leathers and Olson, 2012), including the SC (Ikeda et al., 2003). […] The magnitude of such responses would then reflect the value of the predicted reward."

This is exactly the process we envisage as an explanation of how extensive Pavlovian pre-tuning according to value can influence the magnitude of initial sensory processing in primary sensory structures all over the brain….there are many references, mainly from the attentional literature, indicating that some form of long-term plasticity in primary sensory structures occurs when an arbitrary stimulus is associated with reward or punishment.

Quote from our paper: "There are many papers that demonstrate that an association with, or an expectation of reward can dramatically influence the magnitude of the initial sensory response in primary sensory areas throughout the brain (Mogami and Tanaka, 2006; Serences and Saproo, 2010; Metzger et al., 2006; Leathers and Olson, 2012), including the SC (Ikeda et al., 2003). […] After training it is likely that CSs presented at different locations within the visual field would elicit differential responses in tectonigral neurons. The magnitude of such responses would then reflect the value of the predicted reward."

However, in my view value-related signals in the SC are more likely to emerge due to short-term changes in SC response gain caused by top-down modulatory inputs from other areas (e.g. spatial attention based on reward value). This seems more compatible with the Ikeda study cited here, since the rewarded location changed rapidly during that experiment, which the SC can adapt to in < 3 trials (Isoda and Hikosaka, Journal of Neurophysiology 2008).

This is certainly true for the Ikeda study….there it is very clear that top-down specification of whether the monkey is in a rewarded vs. non-rewarded trial influences the magnitude of the initial sensory response in the SC. However, we are proposing a different process whereby prior Pavlovian association with reward is quite sufficient to modify the initial sensory response in early sensory areas including the SC – to reiterate, there is abundant evidence in the attention literature indicating Pavlovian sensory pre-tuning happens.

If so, then despite the authors' apparent emphasis on bottom-up retino-tecto-nigral transmission, the value signals are actually injected into this pathway by other (possibly cortical) brain areas that send a top-down modulatory signal to the SC to change its response gains at different parts of its visual map.

Given the short latency of the value-coded latency in the DA neurons and the longer latencies of cortical sensory processing or potential value coding of sensory information from the SC, we think this suggestion is unlikely. Indeed, the new data we have included in Figure 3 shows that the earliest time points that value differentiation occurred was not different between responses to CSs in intact and in lesion-affected visual fields.

P.S. I suggest not using the term "sensory pre-conditioning", since it refers to a specific kind of training without reinforcement (Brogden, 1939), which isn't what the authors want to say here.

We removed the sentences including the term.

2) Figure 4 shows the responses to CS/outcome for LR, which look beautiful. But it is also important to show the responses to CS/outcome for SR, even though only the LR CS was presented in the inactivated part of the visual field. This is because the key point is whether discrimination between the two CSs is impaired, not just the response to any single CS. Indeed, this distinction was important in the analysis of behavior: while inactivation abolished one animal's licking to the LR CS, for the other animal it induced equal licking for both CSs. Also, if the authors do see a change in DA responses to the SR CS/outcome despite it being outside the inactivated field, that would be interesting!

We have calculated the averaged response to SR-CS and firing rate in each DA neuron before and after the injection (Figure 4—figure supplement 2). However, presumably because of the contrast effect with the LR-CS, responses to SR-CS were small, and the magnitude of responses was not different before and after the muscimol injection.

[Editors' note: further revisions were requested prior to acceptance, as described below.]

Essential points:

1) The authors still emphasize subjective awareness but this is not very well documented or backed up by data. It is also noted that the lesions might have been made a long time ago (~10 years?).

We added the time from surgery of the V1 lesion to the experiments in this study. They were conducted 46 months (monkey K), 44 months (monkey U), and 6 months (monkey T) before the experiments (start of subsection “V1 lesion”; subsection “Subjects”).

Given these problems, we would like the authors to further address these points. Do the monkeys recover any (conscious) visual abilities (It wouldn't need to be high acuity vision to complete this task)? Was the data in Figure 1—figure supplement 1 about contrast sensitivity recorded recently, or is it adapted from the 2008 paper? In addition to discussing possible recovery, the authors need to be much clearer (throughout the entire paper) about the limitations of this study in the absence of psychophysical tests.

We assessed their visual ability (Figure 1—figure supplement 1) at the starting point of the present experiments. We added this information to the revised manuscript (subsection “V1 lesion”). All the monkeys in this study were different from monkeys in Yoshida et al. in 2008, and we referred only the method part of the paper in 2008. We made this point clearer in the revised manuscript. In our previous paper (Yoshida and Isa, Sci Rep 2015), we assessed visual awareness of the V1 lesioned monkeys using signal detection theory, however, we did not test it on the monkeys in this study and it is not the main issue of this paper. Therefore, we removed the whole descriptions on the visual awareness from this paper (the last paragraph in Discussion).

2) The same problem, regarding phenomenal awareness, is also important for the data in Figure 4 (right, reward response). This data demonstrate that the SC is necessary for visual reward prediction, but it appears impossible to know whether the muscimol inactivation of SC caused loss of phenomenal awareness, or whether the muscimol inactivation was done in the absence of phenomenal awareness and thus constitutes a neural mechanism for unconscious reward processing. The authors suggest the latter (end of subsection “Possible input pathways for reward prediction”), but it does not seem to be an appropriate conclusion (or suggestion) because the authors did not measure conscious awareness.

It is true that we did not assess the visual awareness after inactivation of the SC. As described above, we removed the description on the visual awareness from this paper.

3) Although the authors acknowledge that their paradigm cannot discriminate between salience and value, the title of the paper still reads as value-coding. Because the cues probably had very similar physical properties, and hence similar physical salience, this problem can be dealt with by more careful wording throughout. It appears to be confusing if the authors just add a disclaimer paragraph to the Discussion, and ignoring the disclaimer everywhere else. We suggest that the authors remove "value-coded" and then use more careful wording and make the manuscript consistent throughout the manuscript.

We changed the word ‘value-coded signal’ to ‘visually-evoked reward expectation signal’ in the title, and to similar words in other parts of the text. (But if the words are reference from the previous literature, and were not directly relevant to the present experimental results, we kept them.)

In addition, we made small changes to the text to make the manuscript concise and correct typos.